# Trace CH4 Gas Detection Based on an Integrated Spherical Photoacoustic Cell

Yexiang Jiao [1], Hongji Fan [1], Zhenfeng Gong [1,*], Kai Yang [1], Feiyang Shen [2], Ke Chen [1], Liang Mei [1], Wei Peng [2] and Qingxu Yu [1]

1 School of Optoelectronic Engineering and Instrumentation Science, Dalian University of Technology, Dalian 116024, China; jiaoyexiangmark@mail.dlut.edu.cn (Y.J.); fsfhj20001017@mail.dlut.edu.cn (H.F.); yangkai666@mail.dlut.edu.cn (K.Y.); chenke@dlut.edu.cn (K.C.); meiliang@dlut.edu.cn (L.M.); yuqx@dlut.edu.cn (Q.Y.)

2 School of Physics, Dalian University of Technology, Dalian 116024, China; sfy828@mail.dlut.edu.cn (F.S.); wpeng@dlut.edu.cn (W.P.)

* Correspondence: gongzf@dlut.edu.cn; Tel.: +86-411-84708379

**Abstract:** This paper presents an integrated spherical photoacoustic cell (SPAC) for trace methane ($CH_4$) gas detection. Theoretical analysis and analogue simulations are carried out to analyze the acoustic field distribution of the SPAC at resonant and non-resonant modes. The finite element simulation results based on COMSOL show that the first-order radial resonant frequency and second-order angular resonant frequency are 24,540 Hz and 18,250 Hz, respectively, which show good agreements with the formula analysis results. The integrated SPAC, together with a high-speed spectrometer and a distributed feedback (DFB) laser source, makes up a photoacoustic (PA) spectroscopy (PAS) system, which is employed for $CH_4$ detection. The minimum detection limit (MDL) is measured to be 126.9 parts per billion (ppb) at an average time of 1000 s. The proposed SPAC has an integrated, miniaturized and all-optical structure, which can be used for remote and long-distance trace gas detection.

**Keywords:** photoacoustic spectroscopy; spherical photoacoustic cell; $CH_4$ detection; finite element analysis





## 1. Introduction

Trace gas detection plays an important role in environmental atmospheric monitoring, medical clinical diagnosis and industrial control [1–3]. Photoacoustic (PA) spectroscopy (PAS), which utilizes the PA effect of the gas, is the most common method in trace gas analysis due to its fast reproduction speed, strong selectivity and high sensitivity [4–6]. The PA effect refers to that when the sample in the PA cell is irradiated by a beam of modulated or pulsed monochromatic light, the absorbed light energy is partially or completely converted into heat in a non-radiative relaxation way, causing the heated volume of the sample to expand to produce pressure waves that extend outward with the center of the light source, and the PA signals can be detected by an acoustic sensor placed in the PA cell [7]. The amplitude of the PA signals generated in the PA cell is proportional to the intensity of the incident light, the sensitivity of the acoustic detector and the PA cell constant [8].

In recent years, many researchers have done a lot of researches in order to optimize the PA system. With the development of laser technology, the near-infrared distributed feedback (DFB) diode laser combined with the erbium-doped fiber amplifier (EDFA) were applied as the excitation source to enhance the PA signals [9,10]. In 2013, Ma et al. used a mid-infrared quantum cascade laser (QCL) as the excitation source to achieve a minimum detection limit (MDL) of 340 parts per trillion (ppt) and 4 parts per billion (ppb) for carbon monoxide (CO) and nitrous oxide ($N_2O$) gases under the optimal data acquisition time of 500 s [11]. In order to further improve the amplitude of PA signal, a high-sensitivity quartz

tuning fork (QTF) and a fiber-optic acoustic sensor were used to replace the traditional microphone as the acoustic detector [12–23]. In 2002, Kosterev et al. firstly proposed quartz enhanced photoacoustic spectroscopy (QEPAS) technology. Instead of a gas-filled resonant acoustic cavity, the acoustic energy is accumulated in a high-Q QTF. With the laser output power of 8.44 mW and the working wavelength of 1.53 nm, the MDL of acetylene ($C_2H_2$) gas was 2 parts per million (ppm) [12]. In our previous work, we proposed a low-frequency PA sensor based on a Parylene-C diaphragm for space-limited trace gas detection, the detection limit of $C_2H_2$ was 88.4 ppb with a locking integral time of 1 s [13].

Except for the incident light and the acoustic detector, the PA cell has a great impact on the performance of the PAS system. Zheng et al. designed a compact differential PA cell consisting of two acoustic resonators [3]. The PAS system has achieved a detection limit of 3.6 ppm with an average time of 1 s. In our previous work, a T-type longitudinal resonant PA cell was developed for trace gas detection [24]. The T-type PA cell possesses a high PA cell constant, a simple manufacturing process and a fast response time. However, at present, the PA cell generally adopted a cylindrical structure [3–5,9,13–15,24–37], and there are few studies on other shapes of PA cells. Wang et al. proposed an ellipsoidal resonant PA cell and concluded that the Q value of the ellipsoidal PA cell could reach 70 through theoretical and simulation analyses [38]. There is also some research about spherical PA cells (SPAC) [39–41]. Shi et al. proposed that, through theoretical analysis, it was proved that there was no viscous loss in the radial mode of a spherical PA cell; that is, it had a better Q value than the cylindrical PA cell [39]. Afterwards, Zhao et al. made a more detailed theoretical analysis of the SPAC on this basis [40]. However, the above studies on the SPAC were lacking in relevant simulation analyses to verify the theoretical results. Furthermore, the PAS system based on the SPAC had a large volume, which was not suitable for on-site measurements.

In this paper, an integrated SPAC is designed and manufactured, and the sensitivity and frequency response of the SPAC are simulated in detail using COMSOL. The first-order radial resonance frequency and the second-order angular resonance frequency of the SPAC are determined. The SPAC is applied in an all-fiber PAS system based on the white-light interferometry (WLI) demodulation algorithm. Even though the PAS system operated at a non-resonant mode, the reduction in sensitivity is fully compensated by the high sensitivity of the acoustic sensor working at the resonant frequency, and a high-sensitivity detection of $CH_4$ gas is realized. Compared with the other SPACs presented so far, the proposed integrated SPAC has the characteristic of a small volume, and can be used for remote and long-distance measurement of trace gas detection.

## 2. SPAC Design and Theoretical Analysis

The schematic diagram of the integrated SPAC is shown in Figure 1. The whole device is divided into two parts. The first part of the SPAC includes a hemispheric PA cavity, two vents and two air taps. The second part consists of the other hemispheric PA cavity, a collimator, a ceramic ferrule, a cantilever and two single-mode fibers. The two parts are coupled to each other through an intermediate thread to form the spherical PA cavity. The inner diameter of the PA cell is 20 mm. The outer diameter of the cylindrical brass shell is 30 mm. The single-mode fiber connected with a DBF laser is coupled to the SPAC by a collimator. A ceramic ferrule located in the center of the brass shell is used to transmit the detection light of the fiber-optic cantilever-based acoustic sensor. The length and width of the cantilever beam are 2.8 mm and 1.0 mm. Two small vents with a radius of 0.25 mm, which are connected to two air taps, are opened inside the brass shell for the exchange of measured gas. In order to ensure the air tightness of the device in the experiment, the joint of the two parts is sealed using a silastic sealed ring.

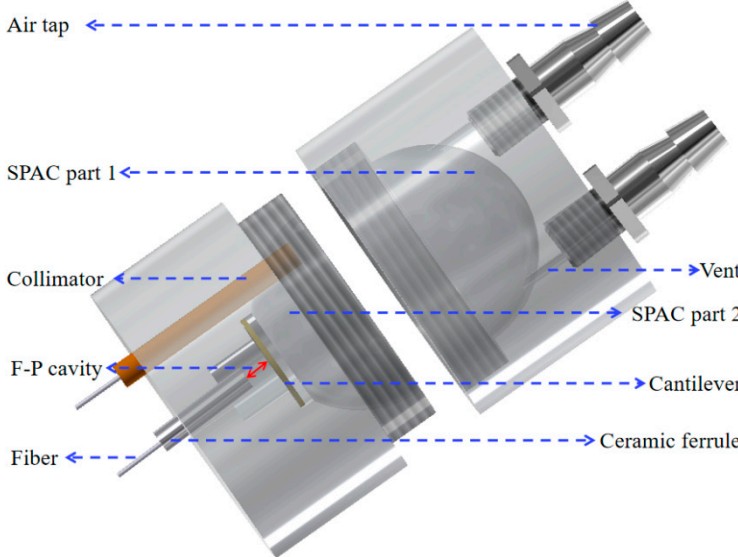

**Figure 1.** The structure of the SPAC.

The frequency response curve of the cantilever-based acoustic sensor is tested based on the WLI demodulation algorithm. As shown in Figure 2, the sensor has the highest sensitivity at a resonant frequency of 1290 Hz, which is the first-order resonant frequency of the cantilever beam.

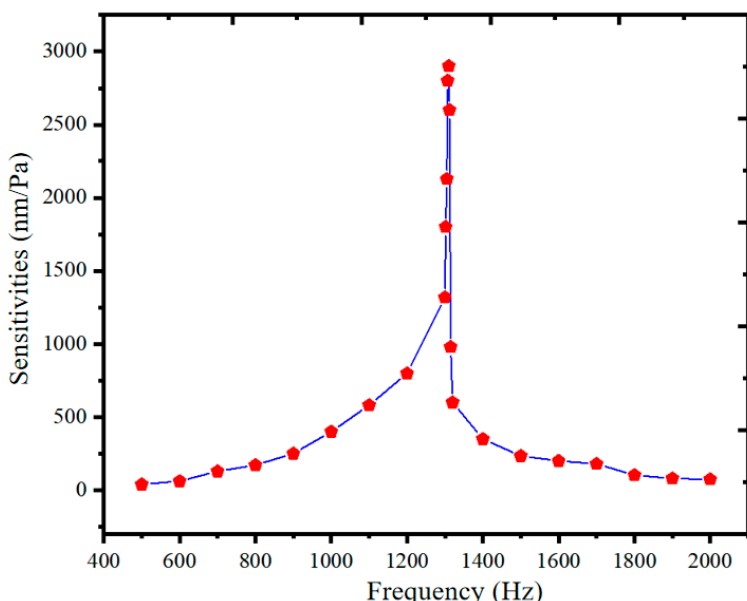

**Figure 2.** Frequency response of the cantilever-based acoustic sensor.

In a PAS system, when the gas molecules absorb light energy transitioning from the ground state to the excited state, part of it is converted into translational energy of the gas, and then expressed as heat energy to increase the temperature of the gas in the cell. Since the volume of the PA cell is fixed, the increased temperature will lead to an increase in pressure, and then acoustic signals will be generated under the action of periodic excitation of the light source. In the case of a weak absorption, the thermal power density generated by the incident light in the gas is $H\left(\vec{r}, t\right) = \tau/\tau_c\left[\beta I\left(\vec{r}, t\right)\right]$, where $\beta$ is the absorption coefficient of the gas, $I(r, t)$ is the intensity distribution of the incident light, $\tau$ is the vibrational relaxation time of gas molecules in the PA cell, $\tau_c$ is the vibration–translational relaxation time of gas molecules in the PA cell, and $H$ has a linear relationship with $\beta$ and $I$.

For a SPAC, the change in the gas acoustic pressure in the cell $p\left(\vec{r}, t\right)$ is described by the acoustic equation [39]:

$$\nabla^2 p\left(\vec{r}, t\right) - \frac{1}{c^2}\frac{\partial^2 p\left(\vec{r}, t\right)}{\partial t^2} = -\frac{\gamma-1}{c^2}\frac{\partial H\left(\vec{r}, t\right)}{\partial t}, \tag{1}$$

where $\vec{r}$ is the position vector, $c$ is the acoustic velocity in the PA cell, $p$ is the acoustic pressure in the cavity, and $\gamma$ is the specific heat capacity ratio (the ratio of specific heat capacity at constant pressure to specific heat capacity at constant volume).

The wave equation satisfies the boundary condition on the cell wall:

$$\frac{\partial p\left(\vec{r}, t\right)}{\partial n} = 0, \tag{2}$$

where $n$ is the normal vector at the boundary of the spherical cavity.

For a standard spherical cavity, the resonant acoustic mode $p_j\left(\vec{r}\right)$ (expressed in spherical coordinates) and the resonant frequency $f_j$ can be solved as:

$$p_j\left(\vec{r}\right) = p_{nlm}(r, \theta, \varphi) = j_l(k_{l,n}r)P_l^m(cos\theta)e^{im\varphi}, \tag{3}$$

$$f_j = k_{l,n}c/2\pi R, \tag{4}$$

where $j = (nlm)$, $n$, $l$, $m$ are the order of radial, theta angle and $\varphi$ angle direction acoustic modes, respectively. $j_l(r)$ is the first-order spherical Bessel function, $P_l^m(x)$ is a joint Legendre function, $k_{l,n}$ is the n-th root to satisfy the boundary conditions, which can be expressed as:

$$\left.\frac{\partial j_l(kr)}{\partial r}\right|_{r=R} = 0, \tag{5}$$

$P_j(r)$ describes the spatial distribution of acoustic pressure in the spherical PA cavity of mode $j$. The Fourier transform of Equation (1) can be obtained:

$$\left(\nabla^2 + \frac{\omega^2}{c^2}\right)P(r, \omega) = \frac{\gamma-1}{c^2}i\omega H(r, \omega), \tag{6}$$

where $\omega$ is the frequency of the modulated excitation light source, and the acoustic pressure changes are expanded according to each vibration mode:

$$P(r, \omega) = \sum_j A_j(\omega)P_j(r), \tag{7}$$

put in Equation (5) to get the expression of $A_j(\omega)$ as:

$$A_j(\omega) = -\frac{(\gamma-1)\beta P_L \int_{V_c} P_j^*(r)gdV}{(\omega_j/Q)V_c}, \tag{8}$$

In Equation (8), $\omega_j$ is the resonant angular frequency, $Q$ is the quality factor, $V_c$ is the volume of the PA cavity, $P_L$ is the optical power of the incident laser, the overlap integral in the equation represents the coupling degree of the light intensity distribution and the normal mode, and $g$ is the light intensity distribution function.

According to Equations (3) and (4), the spatial distribution and resonance frequency of the mode (100) and the mode (120) can be obtained as follows [39]:

$$p_{100}(r, \theta, \varphi) = j_0(kr) = \sin(kr)/kr, \tag{9}$$

$$f_{100} = 0.72c/R, \tag{10}$$

$$p_{120}(r, \theta, \varphi) = \frac{1}{kr} \times \left[ \frac{-3}{kr} \times \cos(kr) + \left( -1 + \frac{3}{k^2 r^2} \right) \times \sin(kr) \right] \times \frac{1}{4} \times (3\cos(2\theta) + 1), \tag{11}$$

$$f_{120} = 0.53c/R \tag{12}$$

From Equation (9), the acoustic wave in mode (100) only has a radial distribution, and is called the first-order radial resonant mode. This mode is characterized by the acoustic wave propagating along the radial direction, with the maximum acoustic pressure at the center of the cell and a node between the center of the cell and the edge. In addition to the component of radial propagation, the acoustic wave in the SPAS also has an angular component. The mode of (120) is called the second-order angular resonant mode. As shown in Equation (11), in the mode (120), the acoustic wave propagates along the sphere at the boundary of the cell. The acoustic pressure in the radial part is zero at the center of the cell and reaches its maximum at the edge. In the angular part, the acoustic pressure reaches its maximum at 0°, 90° and 180°, and there is a node in the middle. For our PA cell, according to Equations (10) and (12), it can be easily calculated that the first-order radial resonant frequency and the second-order angular resonant frequency are 24,480 Hz and 18,020 Hz, respectively.

A 3D model based on the finite element method was created to make an assay of the PA field distributions of the SPAC. In our numerical calculations, the heat source domain feature was used to replace the heat generated by the high-frequency pulsed laser. The user control mesh was selected. The maximum cell size was 0.8 mm, the minimum cell size was 0.375 mm, the maximum cell growth rate was 1.3, the curvature factor was 0.3, and the narrow area resolution was 5. The results converged completely. The acoustic field characteristics inside the PA cell are the key to the sensitivity and the detection limit of the PAS trace gas detection system, so 16,000~28,000 Hz with steps of 10 Hz were taken as the research range of the simulation frequency, which is shown in Figure 3. There are two peaks of 18,250 Hz and 24,540 Hz, which correspond to mode (120) and mode (100), and are in good agreement with the formula analysis results. Figure 4 presents the acoustic pressure distribution of the SPAC at the second-order angular resonance frequency of 18,250 Hz. As can be seen in Figure 4, the acoustic pressure is zero at the center of the cell and reaches its maximum at the edge. In the angular part, the acoustic pressure reaches its maximum at 0°, 90° and 180°, and there is a node in the middle. Figure 5 shows the acoustic pressure distribution of the SPAC at the first-order radial resonant frequency of 24,540 Hz. The acoustic pressure at the center of the cell is the largest at this frequency with a maximum value of $6.62 \times 10^{-3}$ Pa. There is a wave node from the center to the edge of the cell, and the acoustic pressure decreases from the center to the wave node.

Combining the sensitivity curve of the cantilever beam in Figure 2 with the numerical calculation results of COMSOL in Figure 3, it can be seen that the resonant frequency of the SPAC is very high, which exceeds the operating range of the cantilever-based acoustic sensor according to the Nyquist sampling theory [42]. Figure 6 shows the acoustic pressure distribution of the SPAC at a frequency of 1290 Hz, which is the first-order resonant frequency of the cantilever beam. From Figure 6, it can be seen that the PA cavity works at the non-resonant mode and the acoustic pressure inside the SPAC is almost constant. Although the SPAC is in a non-resonant state at a frequency of 1290 Hz, the signal intensity can still reach $10^{-5}$ Pa at two ends of the central axis, where the cantilever is located. The reduction in sensitivity caused by the operating frequency in the non-resonant region of the SPAC being fully compensated by the high sensitivity of the acoustic sensor working at the resonant frequency. Therefore, the working frequency of the SPAC is selected as 1290 Hz in this work.

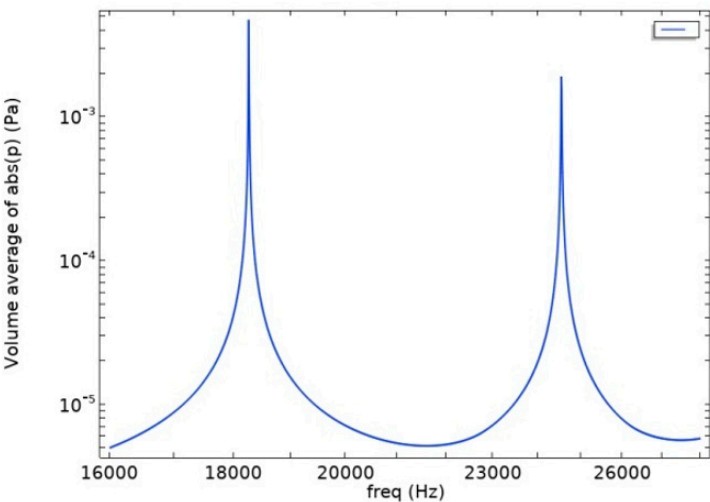

**Figure 3.** Frequency–response curve of the SPAC.

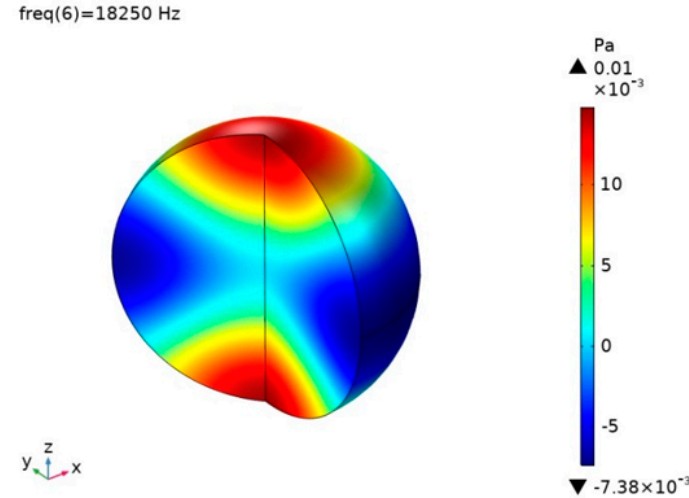

**Figure 4.** Acoustic pressure distribution of the SPAC at the second-order angular resonant frequency of 18,250 Hz.

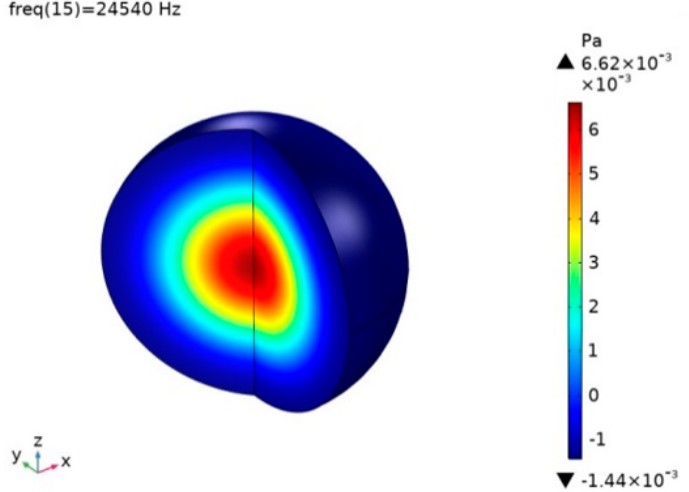

**Figure 5.** Acoustic pressure distribution of the SPAC at the first-order radial resonance frequency of 24,540 Hz.

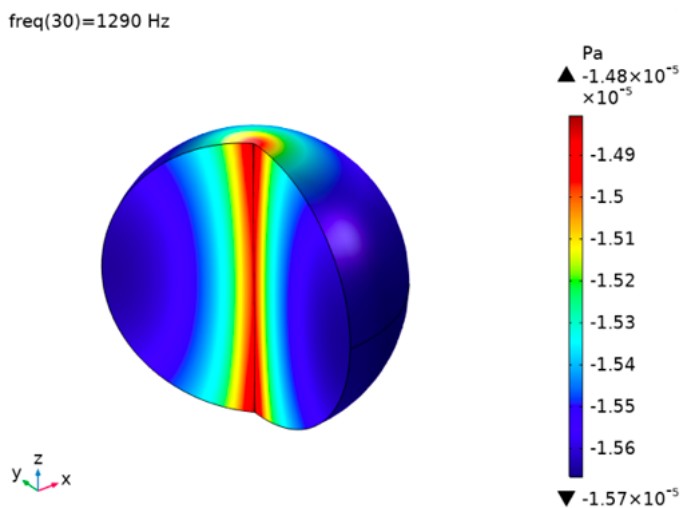

freq(30)=1290 Hz

**Figure 6.** Acoustic pressure distribution of the SPAC at a frequency of 1290 Hz.

## 3. Experimental Results and Discussions

The schematic diagram of the experimental system based on PAS is shown in Figure 7. The system consisted of the SPAC, a near-infrared DFB laser, a high-speed spectrometer, a super luminescent diode (SLD), a circulator, a computer and two mass flow controllers (MFCs). The system used a near-infrared DFB laser with a center wavelength of 1651 nm as the PA excitation source. The laser beam passed through a collimator into the SPAC. The SLD laser was used as the detection source of the fiber-optic acoustic sensor, and the laser was first coupled to the circulator and then propagated into the acoustic sensor. The PA signal deformed the cantilever beam, which in turn caused the F-P (Fabry-Perot) cavity length to change. A LabVIEW-based phase-locked WLI demodulation algorithm based on the high-speed spectrometer was used to demodulate the dynamic cavity length of the acoustic sensor and the magnitude of the gas concentration was obtained. Two MFCs were used to control the concentrations of the $CH_4/N_2$ mixture. Because the PA system operated at a non-resonant mode, which avoided the resonant frequency drift due to the temperature change; meanwhile, the experimental procedure was carried out at a constant room temperature and atmospheric pressure—the temperature sensitivity of the SPAC was negligible.

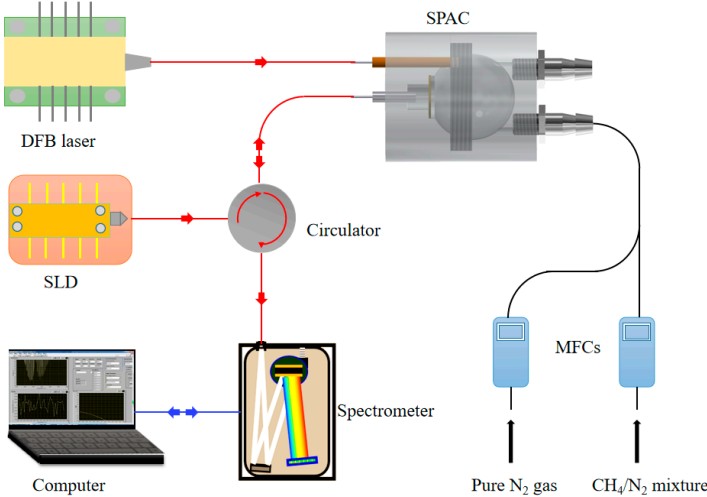

**Figure 7.** The PAS experimental system based on the SPAC.

The two MFCs with $\pm$1% error, together with one bottle $CH_4/N_2$ gas mixture, and one bottle of pure $N_2$ used as standard gas, made up the gas mixing system. Five different concentrations of $CH_4/N_2$ gas mixture from 50 ppm to 1000 ppm were sequentially passed into the SPAC. The PA signals were measured with the PA cell confined. Figure 8 shows the second harmonic responses of $CH_4$ gas with the concentrations of 50 ppm, 100 ppm, 200 ppm, 500 ppm, and 1000 ppm, respectively, and Figure 9 shows the linear results of the second harmonic signals to the concentrations of $CH_4$ gas at the corresponding peak, with a slope value of 0.26 pm/ppm and a fitted R-squared of >0.998, which verifies that this PAS system response shows an approximately linear response to different concentrations of $CH_4$.

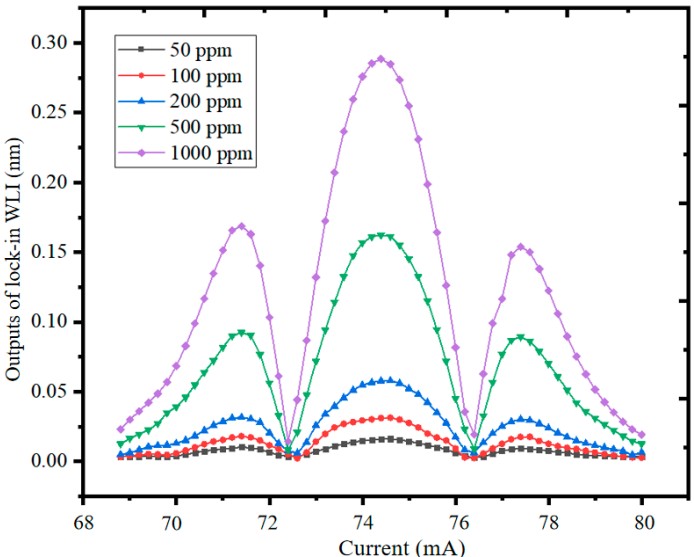

**Figure 8.** The output of lock-in WLI of $CH_4$ gas with different concentrations.

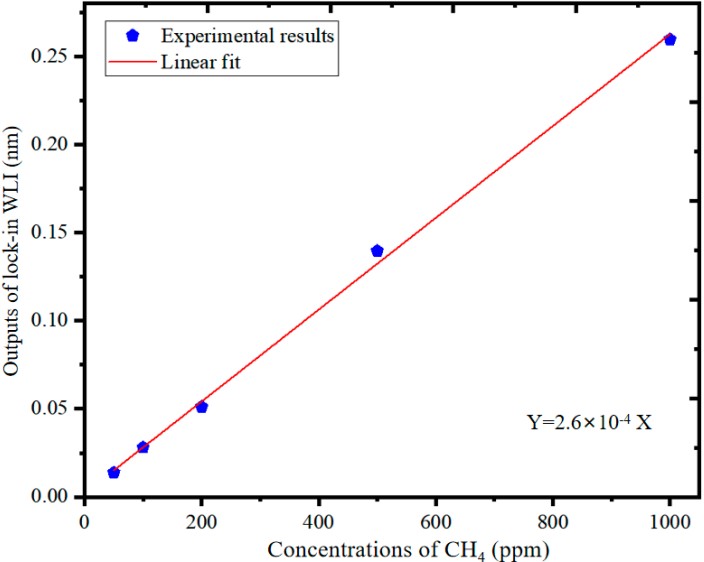

**Figure 9.** The output PA signals changing with different concentrations of $CH_4$.

The background noise of one thousand sample points was obtained by filling the SPAC with pure $N_2$ gas at an integration time of 10 s, as shown in Figure 10. After calculation, the noise level (1$\sigma$) of the background noise was determined to be 0.32 pm. With the sensitivity of 0.26 pm/ppm, the MDL for trace $CH_4$ detection was calculated to be 1.23 ppm. The Allan–Werle deviation was applied when the SPAC was filled with pure $N_2$. As shown

in Figure 11, the Allan variance was 0.033 pm at an average time of 1000 s. Therefore, the MDL was then estimated to be 126.9 ppb.

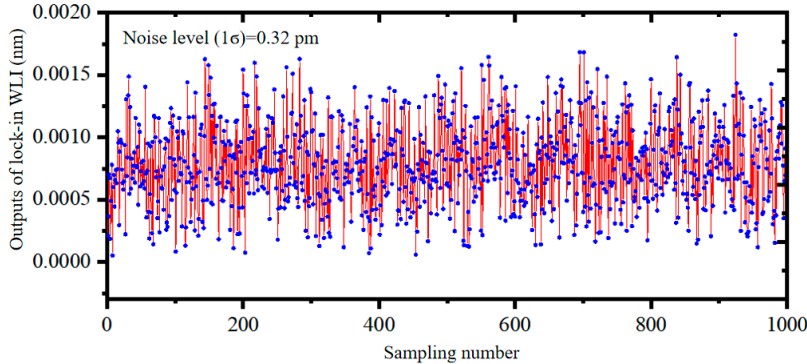

**Figure 10.** The background noises with the SPAC filled with pure $N_2$ gas.

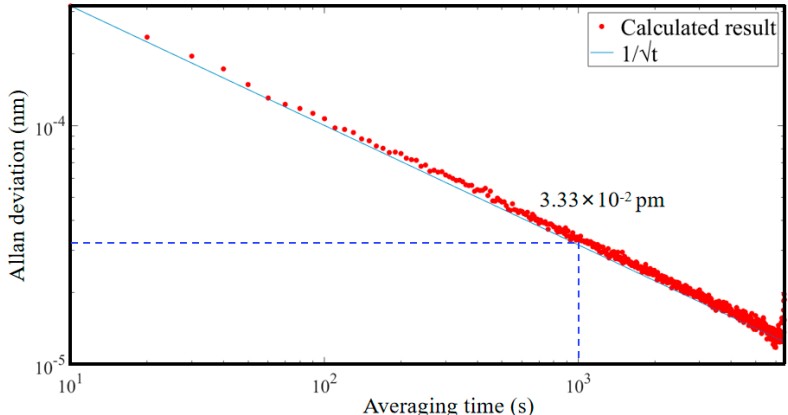

**Figure 11.** The Allan–Werle analysis results of the pure $N_2$ gas.

## 4. Conclusions

In summary, we propose an integrated SPAC for trace $CH_4$ gas detection. The SPAC is divided into two parts, which are coupled to each other through an intermediate thread to form a spherical PA cavity. Formula analysis and numerical calculation are carried out to analyze the acoustic field distribution of the SPAC at resonant and non-resonant modes. From the finite element simulation results, the first-order radial resonant frequency and second-order angular resonant frequency show good agreement with the formula analysis results. A cantilever-based fiber-optic acoustic sensor is used to detect the PA signals based on the WLI phase-locked demodulation algorithm. The integrated SPAC, together with a near-infrared DFB laser and a high-speed spectrometer, makes up a PAS system for trace $CH_4$ gas detection. Considering the sensitivity and the measurement range of the acoustic sensor, the SPAC works at the non-resonant mode. The reduction in sensitivity caused by the operating frequency in the non-resonant region of the SPAC is fully compensated by the high sensitivity of the acoustic sensor working at the resonant frequency. Different concentrations of $CH_4/N_2$ gas mixture are sequentially passed into the SPAC and the experimental results show that the PA signal is proportional to the $CH_4$ concentration. The linearity of the second harmonic response is measured to be 0.26 pm/ppm. The background noise of one thousand sample points is obtained by filling the SPAC with pure $N_2$ gas at an integration time of 10 s. Considering the sensitivity of the SPAC, the MDL for trace $CH_4$ detection is calculated to be 1.23 ppm. The Allan–Werle deviation is applied to make the MDL further decrease to be 126.9 ppb at the average time of 1000 s. Overall, the proposed SPAC has an integrated, miniaturized and all-optical structure, as well as a high

sensitivity for trace $CH_4$ gas detection, which can be used for remote and long-distance trace gas detection.

**Author Contributions:** Conceptualization, Y.J. and Z.G.; data curation, H.F.; formal analysis, Y.J. and F.S.; funding acquisition, Z.G., K.C., L.M., W.P. and Q.Y.; methodology, H.F. and K.Y.; project administration, W.P.; supervision, L.M.; validation, Z.G.; visualization, K.C.; writing—original draft, Y.J. and H.F.; writing—review and editing, Z.G. All authors have read and agreed to the published version of the manuscript.

**Funding:** This research was funded by National Natural Science Foundation of China, grant numbers 11904045, 61905034, 61705031, and 62075025; China Postdoctoral Science Foundation, grant number 2020M673542; Doctoral Start-Up Foundation of Liaoning Province, grant number 2019-BS-051; Natural Science Foundation of Liaoning Province, grant number 2019-MS-054; Dalian High-Level Talent Innovation Support Plan, grant number 2019RQ010; Fundamental Research Funds for the Central Universities, grant number DUT20RC(4)014; State Grid Corporation of China, grant number 521205190014.

**Institutional Review Board Statement:** Not applicable.

**Informed Consent Statement:** Not applicable.

**Data Availability Statement:** Data sharing not applicable.

**Conflicts of Interest:** The authors declare no conflict of interest.

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
