# Peer review of "Trace CH4 Gas Detection Based on an Integrated Spherical Photoacoustic Cell"

_applsci, doi:10.3390/app11114997_

Round 1

Reviewer 1 Report

The authors correctly point out that the signal of spherical resonators operated at a purely radial mode does not suffer from surface losses. But they do not take advantage of this feature since they operate the sensor far off a radial resonance (or any other acoustical resonance) of the resonator. Instead the sensor is operated at the resonance frequency of the cantilever used for signal detection. The authors analytically calculate the resonance frequency for the radial mode. One wonders, why they didn't use the result to match the dimensions of the cantilever to this frequency or adjust the cell radius in order to match the resonance frequencies of cantilever and cell. If they have reasons to reject these options they at least should have discussed them.

In addition the PA signal has been calculated with a FE software. Since in this calculation loss effects have not been taken into account, no benefit of the FE results over the analytical results are obtained.

The FE model is not described in appropriate detail

L104: What is tau, tau_c?

L109: displacement vector -> position vector

L149: were took?

L171: equal -> constant?

L190: Definition of F-P cavity?

Eq. (1): Definition c instead of in L117

Eq. (6): What is v?

Eq. (11): cos(...), sin(...) not in italic

Fig. 2: How has this figure be obtained? Or is it from the data sheet of the cantilever?

Fig. 10: Why are the data points connected by (red) lines?

Conclusion: Contains many details like geometric dimensions etc., which are not appropriate for a conclusion

Incomplete citations:

B Baumann, M Wolff, B Kost, H Groninga, Finite element calculation of photoacoustic signals, Applied Optics, 2007

B. Parvitte, C. Risser, R. Vallon, V. Zeninari, Appl. Phys. B 111(3), 383 (2013)

M Germer, M Wolff, Quantum cascade laser linewidth investigations for high resolution photoacoustic spectroscopy, Applied Optics, 2009

Reviewer 2 Report

The authors present a study on an integrated spherical photoacoustic cell for trace methane gas detection. Theoretical analysis, simulation and experimental work was performed at resonant and non-resonant modes with convincing results. The manuscript is clearly written and concise. I would suggest two (small) points for improvement before publication:

  • please include also the gas flow (was it continuous or interrupted for data recording?) and the temperature (RT?) in the experimental part
  • what about temperature sensitivity? What happens when your sensor is exposed to temperature changes, especially in the resonant case? Here a few sentences should be included into the discussion.

Author Response

Please see the attached file, thanks very much.

Reviewer 3 Report

In my opinion, the manuscript is suitable for publication in Applied Sciences journal after the completion of major revision. 

Therefore, reviewer suggested authors to do revising according to following comments:
1. The Authors should improve the introduction:
- only two papers, which relate to SPAC, are cited in the introduction. Authors should cite other papers relating to SPAC,
- Authors should better describe the novelty of their work. Is presented SPAC described in other scientific papers or offer by some commercial companies? 
- Authors wrote: “The proposed integrated SPAC has the characteristics of small volume, and can be used for remote and long-distance measurement of trace gas detection.” 
Authors should describe devices which were compared with proposed SPAC in area of volume and measurement distance.
2. Authors should improve description of laboratory set-up:
- MFC must be described.
3. In my overall evaluation, the manuscript is poor in terms of discussion. Authors should improve the discussion:
- discussion should relate to both simulation and experimental results. The authors must situate these results into the state-of-art comparing their results with those reported in the literature. The Authors need to highlight the novelty of their work.
4. Authors should improve the conclusions: 
- the conclusions must be substantially processed, because remind discussion. Conclusions should present general claims on the basis of obtained results of simulation and laboratory research. Conclusions should fill research gap which should be described in introduction. 

Author Response

(The authors gave the same response as above.)

Round 2

Reviewer 1 Report

The authors have made changes/corrections to the less important suggestions/errors pointed out by the reviewer. However, concerning the main deficiencies, not much effort for improvement has been made.

1. It still is not clear to me, why a spherical resonator has been used. The advantage of spherical resonators is that is does not suffer from loss when operated in a spherical mode. So if for some reasons it is not possible to take advantage of this, why not using a cylindrical resonator, which is easier to manufacture?

2. When using a FE model it is mandatory to give details concerning the mesh etc. and, in particular, the convergence of the results.

Reviewer 3 Report

I accept in present form.

Author Response

Thanks very much. Please see our revised manuscript.